# Discovery of runs-of-homozygosity diplotype clusters and their associations with diseases in UK Biobank

Ardalan Naseri[1†], Degui Zhi[2]*, Shaojie Zhang[1]*

[1]Department of Computer Science, University of Central Florida, Orlando, United States; [2]Center for Precision Health, School of Biomedical Informatics, The University of Texas Health Science Center at Houston, Houston, United States

**Abstract** Runs-of-homozygosity (ROH) segments, contiguous homozygous regions in a genome were traditionally linked to families and inbred populations. However, a growing literature suggests that ROHs are ubiquitous in outbred populations. Still, most existing genetic studies of ROH in populations are limited to aggregated ROH content across the genome, which does not offer the resolution for mapping causal loci. This limitation is mainly due to a lack of methods for the efficient identification of shared ROH diplotypes. Here, we present a new method, ROH-DICE (runs-of-homozygous diplotype cluster enumerator), to find large ROH diplotype clusters, sufficiently long ROHs shared by a sufficient number of individuals, in large cohorts. ROH-DICE identified over 1 million ROH diplotypes that span over 100 single nucleotide polymorphisms (SNPs) and are shared by more than 100 UK Biobank participants. Moreover, we found significant associations of clustered ROH diplotypes across the genome with various self-reported diseases, with the strongest associations found between the extended human leukocyte antigen (HLA) region and autoimmune disorders. We found an association between a diplotype covering the homeostatic iron regulator (HFE) gene and hemochromatosis, even though the well-known causal SNP was not directly genotyped or imputed. Using a genome-wide scan, we identified a putative association between carriers of an ROH diplotype in chromosome 4 and an increase in mortality among COVID-19 patients (p-value = $1.82 \times 10^{-11}$). In summary, our ROH-DICE method, by calling out large ROH diplotypes in a large outbred population, enables further population genetics into the demographic history of large populations. More importantly, our method enables a new genome-wide mapping approach for finding disease-causing loci with multi-marker recessive effects at a population scale.

## Editor's evaluation

This important study presents a new method for homozygosity mapping in population-scale datasets, based on an innovative computational algorithm that efficiently identifies runs-of-homozygosity (ROH) segments shared by many individuals. Simulation results provided convincing evidence for good accuracy and power of the new algorithm. Application of this new method to the UK Biobank dataset largely recapitulated previously known associations but also revealed a small number of novel discoveries that were missed by existing genome-wide association study methods, highlighting the utility of this new approach. This study will be of substantial interest to readers in human genetics and quantitative genetics.

**\*For correspondence:**
degui.zhi@uth.tmc.edu (DZ);
shzhang@cs.ucf.edu (SZ)

**Present address:** †School of Biomedical Informatics, University of Texas Health Science Center at Houston, Houston, United States

**Competing interest:** The authors declare that no competing interests exist.

## Introduction

Runs-of-homozygosity (ROH) regions are regions of diploid chromosomes where identical-by-descent (IBD) haplotypes are inherited from each parent (*Ceballos et al., 2018*). Traditionally, ROH was thought to be relevant only to inbred populations, and ROH may be linked to consanguinity and population isolation (*Kirin et al., 2010*). However, a growing number of studies of large cohorts and biobanks have found that ROH may be ubiquitously present (*Clark et al., 2019*; *Joshi et al., 2015*). Still, our understanding of the genetic impacts of ROH is limited.

Most existing studies used individuals' global ROH content (the sum of lengths or the count of ROHs) as a surrogate for the degree of inbreeding and associated it with phenotypes. It has long been known that inbreeding is harmful to the health of offspring (*Morton et al., 1956*), and several studies have suggested that the global ROH content is associated with higher risks of recessive disorders (*Lencz et al., 2007*; *Keller et al., 2012*; *Christofidou et al., 2015*). ROHs can also be related to complex traits such as height (*Yang et al., 2010*). With the growing trend of multi-cohort collaboration through meta-analysis, the effect of global ROH content has been studied over very large sample sizes (*Clark et al., 2019*; *Joshi et al., 2015*). A recent study (*Yengo et al., 2019*) revealed that people with extremely long ROH can be found even in outbred populations.

However, collapsing the individual's rich ROH content into a single number summarizing their global content is a drastic oversimplification. In doing so, the opportunities for mapping causal loci of phenotypes are lost. Ideally, one might wish to identify chromosomal regions with a certain ROH diplotype (*Luo et al., 2006*) (pairs of identical haplotypes) and associate the ROH diplotype with the phenotypes of interest. Indeed, homozygosity mapping in pedigree or inbred populations has achieved success in identifying recessive loci (*Keller et al., 2012*; *Lander and Botstein, 1987*; *Leutenegger et al., 2006*; *Pourreza et al., 2020*; *Tischfield et al., 2005*; *Gandin et al., 2015*). However, for general outbred populations, the total number of possible ROH diplotypes at a locus is too enormous to be enumerated efficiently, and ROH mapping of outbred populations has remained only a theoretical possibility.

Here, we proposed an approach, ROH-DICE (runs-of-homozygous diplotype cluster enumerator), that bypasses this impossibly large search space of diplotypes. Instead of enumerating all ROH diplotypes, we focused on those that are sufficiently long and frequent. Such ROH diplotypes are of interest because they are at the extreme of distribution: the chance of ROH is determined by the chance of a pair of mates having IBD, and such chance and also the length of IBD segments will decay quickly in outbred populations, as supported by population genetics theory (*Thompson, 2013*; *Donnelly, 1983*) and real-world data (*Ralph and Coop, 2013*; *Naseri et al., 2019b*). However, little is known about such ROH diplotypes because no existing methods can efficiently find them.

We present an efficient positional Burrows–Wheeler transform (PBWT)-based (*Durbin, 2014*) method to find clusters of identical matches. We apply our method to find clusters of ROH diplotypes in UK Biobank data. Each cluster of ROH diplotypes is defined as a set of 100 consecutive homozygous sites that are shared among over 100 individuals. We investigate the association between the detected ROH diplotype clusters and self-reported non-cancerous diseases and present the results for the disease having the strongest associations with the detected ROH diplotype clusters.

## Results

### Methods overview

An ROH diplotype is a pair of homozygous haplotypes of an individual. A frequent ROH diplotype is one shared by several individuals at the same location and with the same consensus sequence. Although long and frequent ROH diplotypes are not very common, it is difficult to enumerate ROH diplotypes above a certain length and a certain frequency. We refer to frequent ROH diplotypes above a certain frequency (set of individuals) and a length as ROH clusters. As a compromise, ROH regions are traditionally aggregated into single numbers and their association with phenotypes is investigated. As a result, the loci-specific association signals of the ROHs or the allele-specific signals are likely to be lost (see *Figure 1*).

To solve this problem, we first processed the biallelic genotype panel (with three possible values 0, 1, and 2 at each position) by randomly assigning any heterozygous sites to homozygous sites with the reference or the alternative allele. The reasons for such processing are twofold. One, through this conversion, the true ROH diplotype clusters, mostly consisting of homozygous sites, are relatively

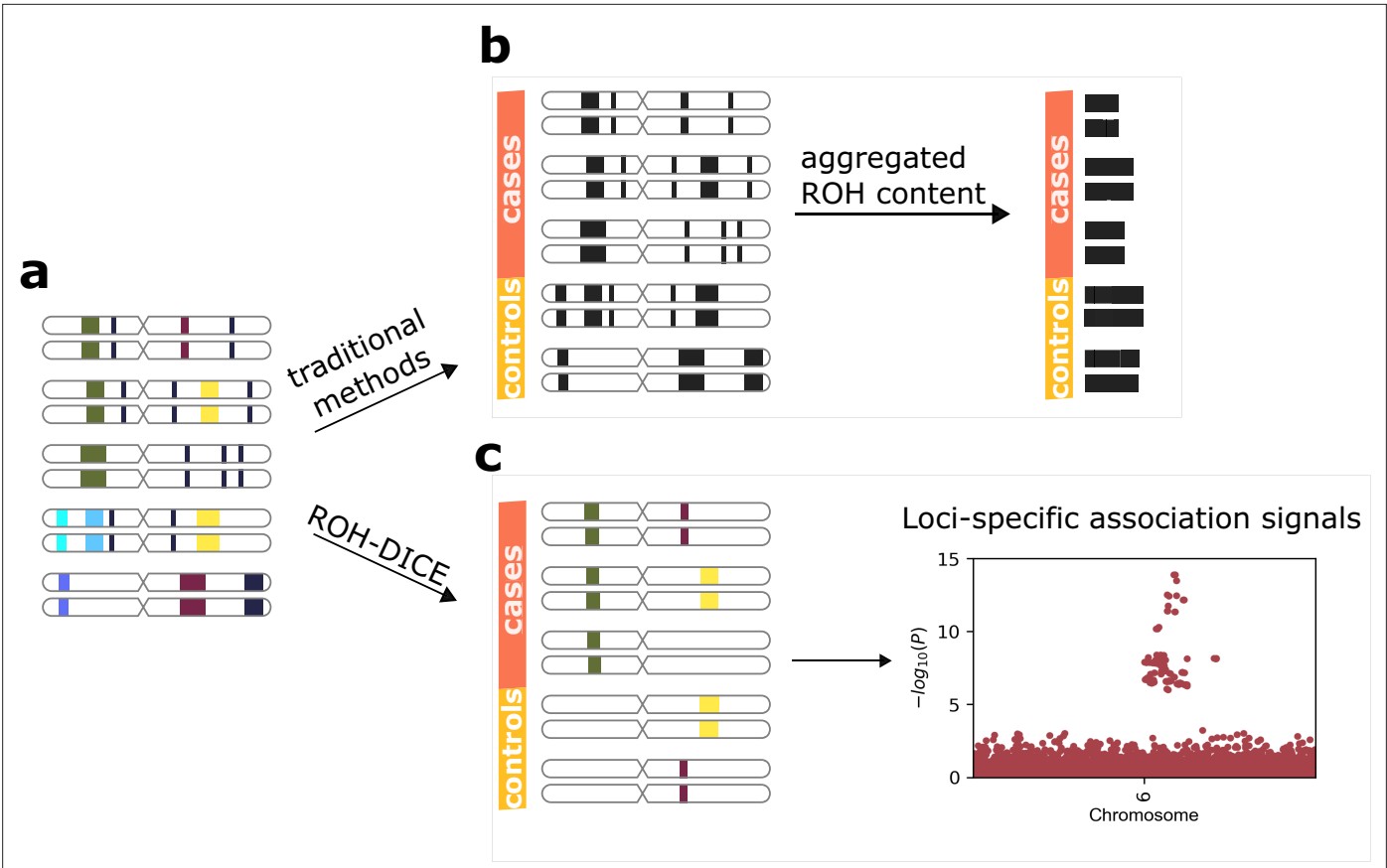

**Figure 1.** Runs-of-homozygosity (ROH)-DICE enables the discovery of loci-specific association signals of ROH diplotypes. The actual ROH contents (**a**) including the locations and sequence identities of ROH (indicated by different colors) were lost in traditional ROH analysis pipelines (**b**) which aggregate the ROH contents per individual and lose the chances for identifying associating loci. ROH-DICE (**c**) reveals ROH diplotype clusters that are long and wide enough, thus enabling mapping loci associated with phenotypes.

The online version of this article includes the following figure supplement(s) for figure 1:

**Figure supplement 1.** Evaluation of runs-of-homozygosity (ROH) clusters using simulated genotype data with and without genotyping errors.

**Figure supplement 2.** Evaluation of runs-of-homozygosity (ROH) clusters using different cut-off values for the same target length (*L*) and width (*W*).

**Figure supplement 3.** Power of runs-of-homozygosity (ROH)-DICE vs traditional genome-wide association studies (GWAS) for finding associations between phenotypes and ROH clusters using 200 samples with 10 Mbps and 100 consecutive causal variant sites.

intact and will still have a high probability of maintaining a good portion of their haplotype. However, some post hoc processing may be needed to merge the ROH diplotype clustered with minor deviations of their consensus sequences. Notably, this conversion should introduce very few false positives as when the length and the width cut-offs are large, there is little chance a non-ROH diplotype cluster will emerge. Two, this effectively converts the panel into a haplotype panel (with two possible values 0 and 2 at each position), where efficient algorithms for identifying haplotype matching blocks are available. A haplotype matching block is defined as a sequence of variant sites that have a predefined minimum frequency. An extra benefit is, by doing this conversion, no phasing of haplotypes is needed.

Haplotype matching blocks can be identified by leveraging the efficient sorting of haplotypes in the PBWT data structure. For a haplotype panel, PBWT sorts haplotype sequences at each variant site according to their reverse suffixes, and thus a set of haplotypes sharing the same sequence before a variant site will be adjacent in the sorting and form a 'matching block'. We use auxiliary PBWT data structures to keep track of the length (the number of variant sites) and the width (the number of haplotype sequences) of the matching block and trigger the output report by watching the data structures. *Figure 2* summarizes the overall ROH-DICE method. More details about the algorithms for finding blocks of matches and searching for ROH diplotypes are presented in the Methods section.

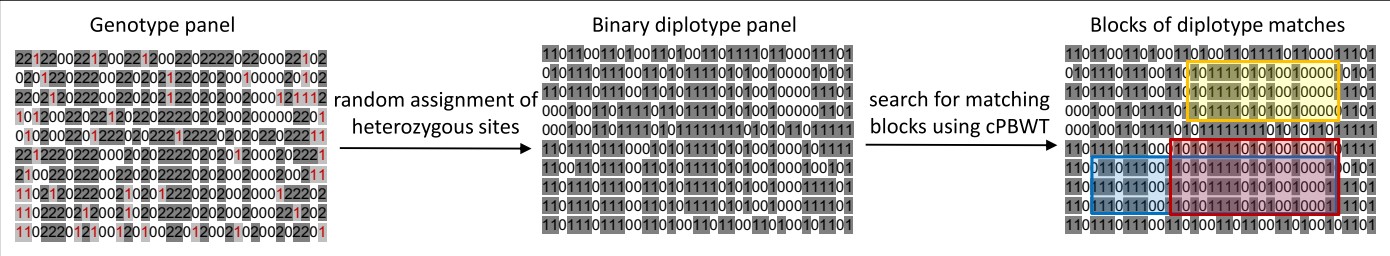

**Figure 2.** A simple schematic of searching for runs-of-homozygosity (ROH) diplotype clusters in a genotype panel. The input is a genotype panel where each line represents an individual. The heterozygous sites are depicted in violet in the genotype panel. Input genotype data are converted into a binarized genotype panel where homozygous sites are preserved. The matching blocks (clusters) are searched using consensus PBWT (cPBWT). A matching block is defined by a minimum number of sites, individuals, and also an objective function. The objective can be either maximizing the number of individuals or maximizing the number of sites. The clusters of matches are highlighted in different colors. Red represents a cluster with the maximized number of individuals and blue represents a cluster with the maximized number of sites.

## Evaluation of ROH clusters in simulated data

### Accuracy and power of ROH clusters

To evaluate the detection power and accuracy of ROH clusters, we simulated 200 individuals of European ancestry using msprime (*Baumdicker et al., 2022*). The IBD segments were computed using the *tskit* (*Kelleher et al., 2018*). The *tskit* package extracts IBD segments between any two individuals and genomic loci where the alleles have been inherited along the same genealogical path. The variant sites with a minor allele frequency of less than 1% were filtered out. We then created an artificial variant call file where the number of sites corresponds to the number of sites in the original file. We iterated over each pairwise IBD segment and assigned the identical randomly selected alleles for all sites covering the IBD segment. Finally, we ran a modified version of cPBWT on the interim panel where only homozygous sites are included in each matching block.

We extracted the cluster from the ground truth with the maximum overlap for each reported cluster and computed the overlap ratio. The accuracy is then defined as the average of the overlap ratios. The computed accuracy would ensure that a reported percentage of each cluster belongs to the one exact cluster in the ground truth. The power is defined as the average cumulative overlap ratios between the reported clusters and clusters from the ground truth. Large clusters may be reported as two or several smaller clusters due to the strict cut-off values for $L$ and $W$, and the power would determine what percentage of the clusters could be recovered based on the cut-off values. We computed accuracy and power for the reported ROH clusters using haplotypes with 0% and 0.1% genotyping error rates for different $L$ and $W$ cut-off values (*Figure 1—figure supplement 1*). The results show that our approach is robust against genotyping errors up to 0.1%. The detection power for $W = 5$ and $L = 50$ without any error was 79.6% whereas the power for the data with the genotyping error was 79.1%. The accuracy increases with increasing the target lengths and widths. For example for $W = 5$ and $L = 50$, the accuracy was 55% whereas the accuracy for $W = 20$ and $L = 100$ was 63%. *Figure 1—figure supplement 2* shows the detection power for clusters with $L = 100$ and $W = 20$, where different cut-off values were used. The figure shows that the detection power increases with smaller cut-off values. The power for the target values $W = 20$ and $L = 100$ was 34% if the cut-offs were set the same, however, the power increased to 84% with smaller cut-offs ($W = 5$ and $L = 50$). To estimate the power and accuracy for $W = 100$ and $L = 100$, we simulated another dataset containing 1000 individuals of European ancestry with a genotyping error rate of 0.1%. The simulation parameters for this dataset were the same as for the 200 samples (except the number of samples) and ground truth clusters were extracted similarly. The power was 55.96% and accuracy 52.84%, while 58.97% of the reported clusters overlap 50% or more with a ground truth cluster.

### Power of ROH-DICE in association studies

To evaluate the effectiveness of ROH-DICE in association studies, we used the ROH clusters obtained from a sample of 200 genomes of 10 Mbp. We set the minimum length of variant sites to 100 and the minimum number of samples to 5 ($L = 100$ and $W = 5$). We generated 100 phenotypes associated with an ROH cluster for each effect size ranging from 0 to 0.3, using the formula $Y_i = X_i\beta + N(0, \sigma^2)$ with $\sigma^2$

= 0.1. We choose large effect sizes so that the power can be evaluated even with small sample sizes. Here, $X_i$ equals 1 if the sample belongs to the ROH cluster and 0 otherwise.

The total number of variant sites was 23,566, and we extracted 1263 ROH clusters. We calculated the p-values for both ROH clusters and all variant sites. We used a p-value cut-off of 0.05 divided by the number of tests for each phenotype to determine whether the calculated p-value was smaller than the threshold, indicating an association. For genome-wide association studies (GWAS), only one variant site within the ROH cluster, contributing to the phenotype, was required. We tested for all additive, dominant, and recessive effects (*Figure 1—figure supplement 3*). The figure demonstrates that ROH-DICE outperforms GWAS when a phenotype is associated with a set of consecutive homozygous sites. The maximum effect size of 0.3 resulted in ROH clusters achieving a power of 100%, whereas the additive model only achieved 11%, and the dominant and recessive models achieved 52% and 70%, respectively. The GWAS with recessive effect yields the best results among other GWAS tests, however, its power is still lower than using ROH clusters.

## ROH diplotypes in UK Biobank

Here, we searched for the clusters of ROH regions in the UK Biobank data (*Bycroft et al., 2018*). All autosomal chromosomes of all UK participants (487,409) were searched for ROH regions that are shared among at least 100 individuals comprising at least 100 consecutive sites. 56,972 people with self-reported non-British ethnicity in UK Biobank were filtered out. We chose a minimal number of markers that is large enough to avoid an extensive number of clusters. Moreover, the longer the ROH segment, the more likely it is due to shared ancestry rather than statistical noise. Our objective is also to select clusters with a sufficiently large number of individuals to correlate them with phenotypes. It is worth noting that in previous studies, a minimum cut-off of 100 individuals was commonly used (*Lencz et al., 2007*; *Christofidou et al., 2015*; *Moreno-Grau et al., 2021*). On average ~18% of sites are heterozygous, and thus for a pair of 100 sites genotype sequences, there is a very small probability that they will be mapped to the same compressed haplotype. Thus, the rate of false positives should be low. To increase statistical power for downstream association tasks, the width-maximal blocks were reported. This was achieved by running the ROH-DICE program, with a wall clock time of 18 hr and 54 min where the program was executed for all chromosomes in parallel (total CPU hours of ~242.5 hr). The maximum residence size for each chromosome was approximately 180 MB. After running the ROH-DICE program, further post-processing steps were conducted. Each individual with more than 1% heterozygous sites within the block was removed from the cluster. Any two clusters with the same consensus and the exact starting and ending positions were merged.

A total of 1,880,826 ROH clusters (shared among at least 100 individuals and extending at least 100 consecutive sites) were identified in all 22 autosomal chromosomes (*Supplementary file 1*). The average length of these ROHs is 553,095 bp (~0.55 cM). The distribution of ROH clusters is very uneven (*Figure 3a*). Interestingly, the number of ROH clusters in chromosome 6 is the highest. This is mainly due to the excessive number of ROH clusters in the MHC region (65,458). *Figure 4* illustrates the genome-wide coverage of the ROH clusters, with visible peaks at chromosomes 2, 6, and 8. A peak region in chromosome 2 (chr2:135755899–136827560) has been reported to harbor a high selection signal (*Browning and Browning, 2020*). This region contains the lactase gene (LCT) gene which includes a variant selected for lactose tolerance in the European population (*Itan et al., 2009*), though the current understanding of the selection pressure is more nuanced (*Mathieson and Terhorst, 2022*; *Evershed et al., 2022*; *Le et al., 2022*). The most prominent peak in chromosome 6 is located in the MHC region (chr6:28477797–33448354), whose details are shown in *Figure 3b*. The peak in chromosome 8 (chr8:42531565–42629520) contains two known genes, CHRNB3 and CHRNA6. Previous studies have demonstrated the significant role of the CHRNB3–CHRNA6 gene cluster on chromosome 8 in nicotine dependence (*Wen et al., 2016*). Additionally, an earlier study has identified strong evidence for selection in the CHRNB3–CHRNA6 region (*Sadler et al., 2015*). Surprisingly, some clusters comprise more than a hundred thousand individuals sharing the same ROH consensus. The high rate of ROH clusters in the MHC region may be attributed to the high density of markers and low recombination rates (*Traherne, 2008*; *Lam et al., 2013*). We also filtered out all ROH clusters shorter than 0.1 cM (*Figure 3—figure supplement 1*). There is no excessive number of ROH clusters in chromosome 6, as identified by a minimum number of variant sites. The number of samples in ROH clusters within the MHC regions reduces significantly. Although there is still a peak, it is comparable

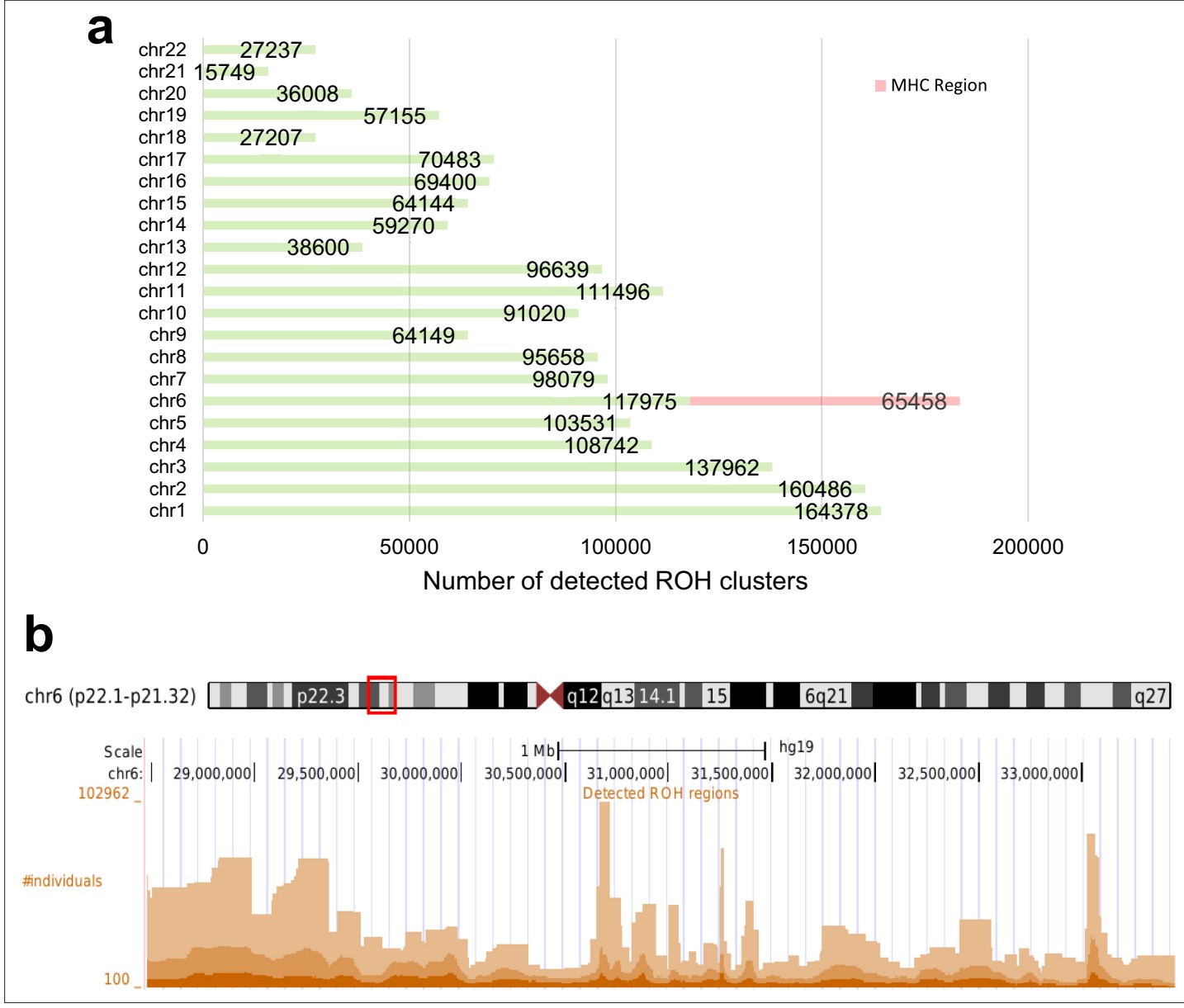

**Figure 3.** Total number of detected runs-of-homozygosity (ROH) diplotype clusters in each autosomal chromosome (**a**) and the detected ROH clusters in the major histocompatibility complex (MHC) region (chr6:28477797–33448354) (**b**) in hg19. Some regions may contain multiple overlapping clusters comprising different sets of individuals. The minimum length of the ROH regions was set to 100 sites and the minimum number of individuals to 100.

The online version of this article includes the following figure supplement(s) for figure 3:

**Figure supplement 1.** Total number of detected runs-of-homozygosity (ROH) diplotype clusters in each autosomal chromosome in UK Biobank with a minimum length (*L*) of 100 sites, a minimum genetic length of 0.1 cM, and a minimum width (*W*) of 100 samples.

to other chromosomes such as chromosome 10 or 12 (*Figure 4—figure supplement 1*). In all subsequent results, we have included clusters with more than 100 sites. However, all the corresponding tables contain the genetic length of the clusters. Low recombination rate regions may contain excessive ROH clusters that we prefer not to discard since it will artificially ignore some ROH clusters driven by selection. The ROH clusters are abundant in regions with low recombination rates and also their distribution is expected to be population specific. Moreover, the 'hotspots' and 'coldspots' may vary in different populations (*Pemberton et al., 2012*). 'ROH hotspots' in study (*Pemberton et al., 2012*) refer to locations where the single nucleotide polymorphism (SNP)-wise ROH frequency is the 99.5th percentile among all frequencies, where a frequency was defined for each variant site. To enable a

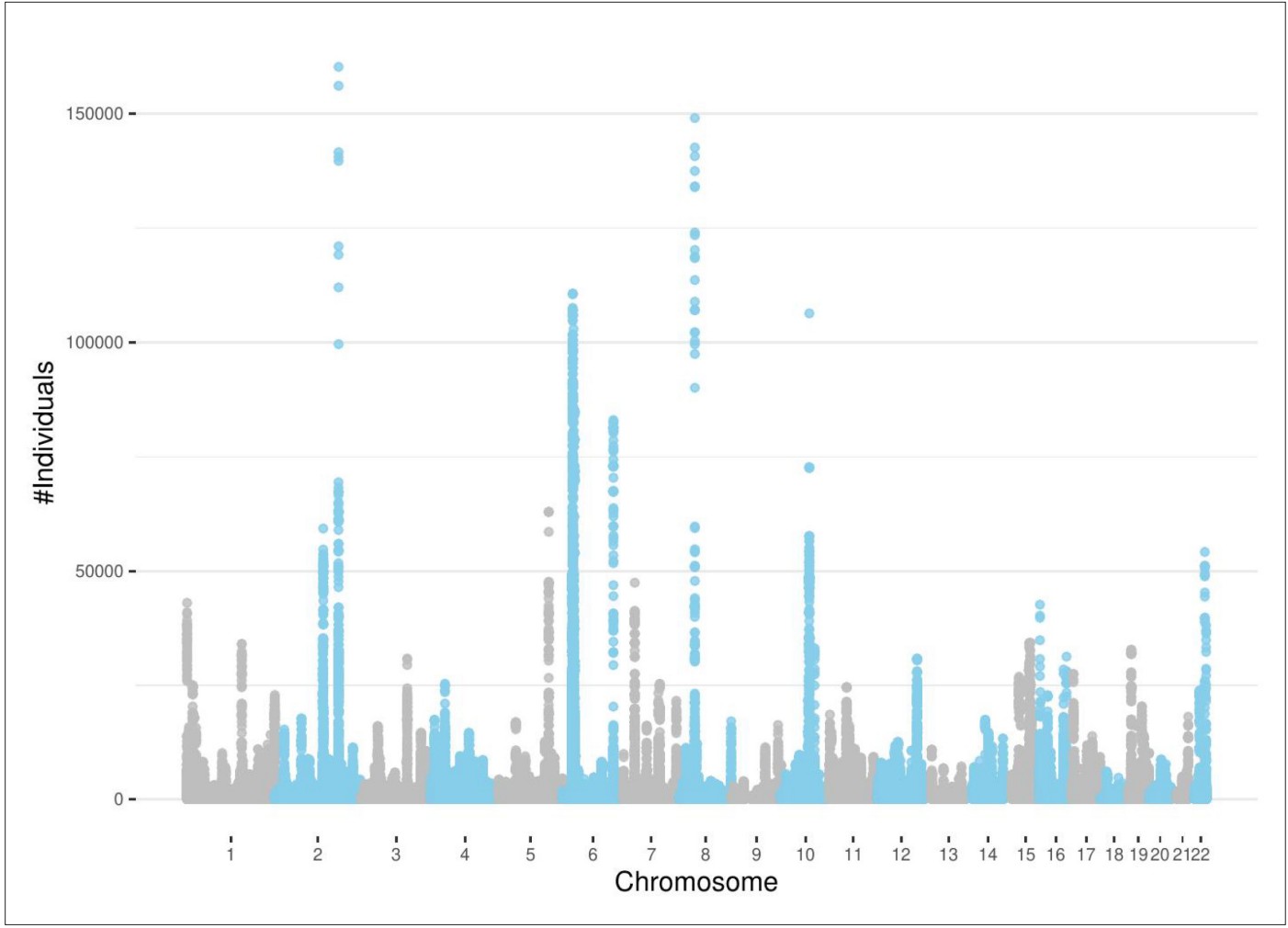

**Figure 4.** Detected runs-of-homozygosity (ROH) diplotype clusters with at least 100 individuals sharing the same consensus with a minimum number of 100 SNPs. Chromosome 18 has the lowest peak for individuals sharing an ROH diplotype. Chromosomes 2, 6, and 8 contain diplotypes shared with more than 100,000 individuals.

The online version of this article includes the following figure supplement(s) for figure 4:

**Figure supplement 1.** The number of individuals sharing the same runs-of-homozygosity (ROH) consensus with a minimum number of 100 SNPs after filtering out segments shorter than 0.1 cM.

comparison with the ROH frequencies from the *Pemberton et al., 2012* study, we also calculated a score for the variant sites by using the intersecting ROH clusters with the sites. We extracted ROH clusters with more individuals than the 99.5th percentile and lower than the 0.5th percentile (see Methods section). The top-ranked ROH 'coldspot' in the European population is located in chromosome 18 (*Pemberton et al., 2012*) and is also identified as below the 0.5th percentile using our method. The top-ranked 'hotspot' was reported in chromosome 15 for Europeans (*Pemberton et al., 2012*) which also overlaps with a peak for British people in chromosome 15 (72100881–72681976) in our study where the number of samples in detected ROH cluster exceeds the 99.5th percentile. The common hotspots and coldspots are listed in *Supplementary files 2 and 3*, respectively. However, further investigation may be required to confirm 'hotspots' as other factors such as marker density may contribute to excessive clusters in certain regions. We also calculated Spearman's rank correlation coefficient ($\rho$) between the two datasets. The correlation coefficient between combined ROH classes in the European population (*Pemberton et al., 2012*) and the ROH clusters in UKBB was 0.54. Of note, *Pemberton et al., 2012* defined three types of ROH clusters (short or class A, intermediate or class B, and long or class C). Our reported ROH regions are based on shared diplotypes with at least

**Table 1.** Clusters of the runs-of-homozygosity (ROH) diplotypes with the lowest p-values in the HLA region for self-reported diseases using the British population in UK Biobank.

Detailed diplotype consensus sequences are available in *Supplementary file 5*. The p-values were calculated using PHESANT. Only the region with the lowest p-value has been included for each disease. Beta represents the effect size reported by PHESANT and *D′* describes the non-random association of an ROH cluster and the overlapping SNP.

| Disease (binary trait) | Diplotype ID | Position (on chr6) | p-value | Beta | Carrier frequency (%) | Odds ratio | Genetic length (cM) | GWAS p-value* | GWAS beta* | GWAS lead SNP* | *D′* |
|---|---|---|---|---|---|---|---|---|---|---|---|
| Ankylosing spondylitis | 1 | 31431031–31464050 | $4.62 \times 10^{-34}$ | 0.121 | 0.29 | 8.66 | 0.071198 | 0 | $1.45 \times 10^{-2}$ | rs113340460 | 0.61 |
| Hemochromatosis | 2 | 25969631–26108168 | $8.02 \times 10^{-120}$ | 0.417 | 0.09 | 24.51 | 0.011597 | - | - | - | - |
| Malabsorption/coeliac disease | 3 | 32564985–32629755 | $3.41 \times 10^{-259}$ | 0.315 | 4.12 | 1.64 | 0.005408 | 0 | $7.74 \times 10^{-3}$ | rs9271352 | 1 |
| Multiple sclerosis | 4 | 32410215–32554129 | $4.36 \times 10^{-45}$ | 0.192 | 0.37 | 3.79 | 0.012736 | $1.05 \times 10^{-107}$ | $4.58 \times 10^{-3}$ | rs9268925 | 0.99 |
| Polymyalgia rheumatica | 5 | 31710968–31794592 | $7.31 \times 10^{-09}$ | 0.080 | 0.23 | 5.90 | 0.006808 | $1.59 \times 10^{-08}$ | $6.80 \times 10^{-3}$ | rs1150748 | 1 |
| Prostate problem (not cancer) | 6 | 34607958–35163974 | $2.84 \times 10^{-08}$ | 0.082 | 0.18 | 6.94 | 0.034889 | $9.81 \times 10^{-04}$ | $9.81 \times 10^{-04}$ | rs76117834 | 0.03 |
| Psoriasis | 7 | 31254263–31263216 | $1.20 \times 10^{-122}$ | 0.214 | 1.21 | 2.73 | $3.07 \times 10^{-05}$ | 0 | $1.93 \times 10^{-2}$ | rs13214872 | 1 |
| Psoriatic arthropathy | 8 | 33072522–33115762 | $8.54 \times 10^{-12}$ | 0.122 | 0.20 | 3.97 | 0.008708 | $4.76 \times 10^{-10}$ | $1.01 \times 10^{-3}$ | rs17221401 | 1 |
| Rheumatoid arthritis | 9 | 32412539–32573760 | $8.15 \times 10^{-122}$ | 0.208 | 0.23 | 2.34 | 0.01293 | $6.96 \times 10^{-124}$ | $8.24 \times 10^{-3}$ | rs188575117 | 0.98 |

*p-values are for the reported SNP from http://www.nealelab.is/blog/2017/9/15/heritability-of-2000-traits-and-disorders-in-the-uk-biobank.

100 SNPs. These regions may not necessarily align with all ROH classes, as variations in length and consensus may lead to differences in ROH regions.

## ROH clusters and disease association

We conducted a phenotypic association analysis of the found ROH diplotype clusters with 445 self-reported non-cancerous diseases, as they are conveniently available in the UK Biobank. We first conducted a quick chi-squared test associating each of the 1,880,826 ROH diplotype cluster membership against each of the 445 phenotypes (see Methods section). The p-values for the 100 regions with the lowest p-values were re-computed using age, sex, genetics principal components, and genotype measurement batch fields by PHESANT (*Millard et al., 2018*) (details see Methods). This identified 61 associations passing the Bonferroni-corrected p-value threshold of $10^{-12}$. *Table 1* shows the p-values for disease associated with the HLA region (chr6) computed by PHESANT. p-values for some diseases are very low in both the chi-squared test and *regression* analysis using PHESANT. It also includes the SNP with the lowest p-value in each cluster that is associated with the corresponding disease. The SNP with the lowest p-value in each cluster was extracted from Neale's lab results [http://www.nealelab.is/blog/2017/9/15/heritability-of-2000-traits-and-disorders-in-the-uk-biobank]. Most of the clusters with low p-values contain at least one SNP with a very low p-value that is associated with the corresponding disease. The top 100 diplotypes with the lowest p-values using chi-squared tests and PHESANT are included in *Supplementary file 4*.

Not surprisingly, the most prominent associations we found are ROH diplotypes in the HLA region with autoimmune diseases. We found that *malabsorption/coeliac disease*, *psoriasis*, *rheumatoid arthritis*, and *multiple sclerosis* have the strongest association with loci in the HLA region. These results are largely consistent with known literature (*Dieli-Crimi et al., 2015*; *Gutierrez-Achury et al., 2015*; *Kurkó et al., 2013*; *Bhalerao and Bowcock, 1998*; *Baranzini and Oksenberg, 2017*; *Canela-Xandri et al., 2018*). One of the most significant associations we identified is the association between the ROH diplotype at chr6:25988167–26122453 and hemochromatosis (p-value = $9.16 \times 10^{-120}$). The frequency of the ROH diplotype is only 0.02% and the odds ratio of having the disease for the carrier

**Table 2.** Clusters of the runs-of-homozygosity (ROH) diplotypes with the lowest p-values outside of the HLA region for self-reported diseases using the British population in UK Biobank.

The p-values were calculated using PHESANT.

| Disease (binary trait) | Diplotype ID | Position | p-value | Beta | Carrier frequency (%) | Odds ratio | Genetic length (cM) | GWAS p-value* | GWAS beta* | GWAS lead SNP* | D′ |
|---|---|---|---|---|---|---|---|---|---|---|---|
| Deep venous thrombosis (dvt) | 10 | chr1:169075589–169528830 | $3.10 \times 10^{-21}$ | 0.039 | 2.08 | 10.49 | 0.56 | $7.41 \times 10^{-166}$ | $-3.13 \times 10^{-2}$ | rs6025 | 1 |
| | 11 | chr1:151515188–151902494 | $1.52 \times 10^{-27}$ | 0.044 | 2.85 | 7.31 | 0.36 | $3.45 \times 10^{-36}$ | $1.43 \times 10^{-2}$ | rs55875222 | 1 |
| | 12 | chr1:151940401–152280032 | $9.46 \times 10^{-24}$ | 0.053 | 11.76 | 2.07 | 0.12 | $1.35 \times 10^{-64}$ | $1.84 \times 10^{-2}$ | rs61815559 | 1 |
| Eczema/dermatitis | 13 | chr1:152493154–152964479 | $1.53 \times 10^{-21}$ | 0.039 | 2.85 | 7.35 | 0.36 | $1.01 \times 10^{-42}$ | $1.62 \times 10^{-2}$ | rs61813875 | 1 |
| Hypothyroidism/ myxoedema | 14 | chr12:111910219–112874179 | $4.51 \times 10^{-21}$ | 0.062 | 5.06 | 1.25 | 0.04 | $1.88 \times 10^{-80}$ | $9.87 \times 10^{-3}$ | rs7137828 | 0.99 |

*p-values are for the reported SNP from http://www.nealelab.is/blog/2017/9/15/heritability-of-2000-traits-and-disorders-in-the-uk-biobank.

is 102.21. Interestingly, several other ROH diplotypes at this locus also have a strong association with hemochromatosis (*Table 1*). This locus is in the extended HLA region and has a low recombination rate. *Hemochromatosis* is an inherited disorder in which iron levels in the body slowly build up over several years. The gene HFE (chr6:26087509–26095469) is a well-known recessive locus for this disease (*Pietrangelo, 2010*). The C282Y polymorphism (rs1800562, chr6:26092913) in HFE is the most penetrant but other polymorphisms with lesser penetrance are also known. Interestingly, the minor allele frequency of the SNP rs1800562 is 6% in the European population but it is not genotyped (and is also not available in the imputed panel) in the UK Biobank data. As a result, this association signal has been completely missing in the Neale Lab results. In another study, the SNP has been imputed and a specific association study for the recessive effect between the homozygous alleles of rs1800562 and hemochromatosis has been reported (*Tamosauskaite et al., 2019*). Our approach found this recessive association signal without direct genotyping of any SNP with high linkage disequilibrium (LD) to the causal SNP, demonstrating the power of our approach beyond regular additive effect GWAS. However, we did not verify that this SNP is indeed part of the ROH diplotype as we do not have access to the WGS data.

We also found some loci outside of the HLA region that are presumably associated with non-cancerous diseases (*Table 2*). The most prominent one is an ROH diplotype at chr1:151515188–151902494 with eczema/dermatitis. This signal overlaps with the GWAS finding of rs4845604 at chr1:151829204 (*Johansson et al., 2019*).

The beta values for effect size were included in all reported tables. These beta values for ROH-DICE are positive, indicating that carriers of these ROH diplotypes may have an increased risk of certain non-cancerous diseases. We also used D′ as a measure of linkage between the reported GWAS results and ROH clusters (see Methods section). We found that most of the GWAS results and ROH clusters are strongly correlated. However, in a few cases, D′ is small or close to zero. In such cases, the reported p-value from GWAS was also insignificant, while the ROH cluster indicated a significant association (See *Table 1* and *Supplementary file 4*). The SNP IDs and consensus alleles for all ROH clusters in *Tables 1 and 2* are reported in *Supplementary file 5*.

## ROH clusters and COVID-19 association

We computed the p-value using the chi-square test for the association between mortality of COVID-19 and the detected ROH regions. We considered only the clusters that had at least 10 cases (tested positive and passed away in 2020). *Figure 5* shows the Manhattan plot for ROH regions and mortality of COVID-19. The most significant ROH region is located in chr4:106318456–106483898 (0.114 cM) with the p-value $1.63 \times 10^{-10}$. 4389 individuals share the diplotypes and 76 of them have tested positive for COVID-19. Eleven persons who carried the same ROH consensus and had tested positive, died in 2020. In other words, carriers of this diplotype have a fivefold mortality compared to non-carriers among COVID-19 patients. We used the GMMAT (*Hoare, 1961*) mixed model regression

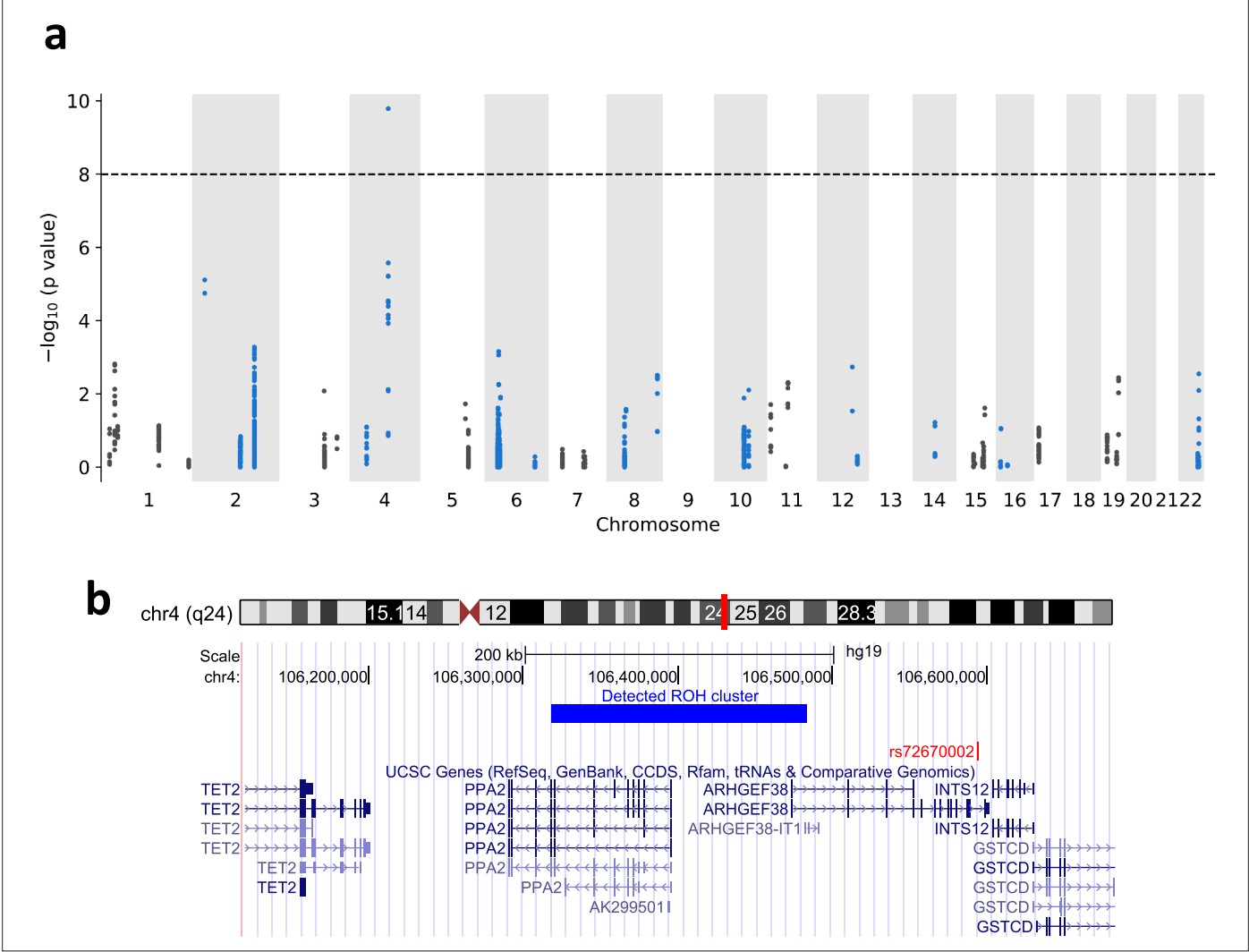

**Figure 5.** Runs-of-homozygosity (ROH) associations between ROH diplotypes and mortality of COVID-19. (**a**) Manhattan plot of ROH diplotypes across all chromosomes and mortality of COVID-19. Diplotypes with less than 10 cases were discarded. (**b**) UCSC genome browser (https://genome.ucsc.edu) view of the region containing the diplotype with a significant p-value in chromosome 4.

to validate the association of this diplotype while adjusting for age, gender, and genetic similarity (see Methods section). The reported p-value was $1.82 \times 10^{-11}$ which is even smaller than the p-value from the chi-square test. The region includes the PPA2 gene. The gene product is an inorganic pyro-phosphatase located in the mitochondrion (*Curbo et al., 2006*). Missense mutations in this gene are reported to cause sudden unexpected cardiac arrest in infancy (*Guimier et al., 2016*). The PPA2 gene has also been recently implicated in COVID-19 through an integrated analysis of GWAS of European patients and lung expressed quantitative trait loci (eQTL) data by the summary-data-based (SMR) method (*Zong and Li, 2021*). The identified region linked to COVID-19 mortality overlaps also with the ARHGEF38 gene. A genetic variant within the gene (rs72670002) has been reported to be signifi-cantly associated with severe illness from COVID-19 in a recent study that used 24,202 cases of critical COVID-19 (*Pairo-Castineira et al., 2023*). Other nearby genes within a 200-kb range include TET2, INTS12, and GSTCD.

## Discussion

In this work, we introduced an efficient algorithm, ROH-DICE, for finding clusters of ROH regions in very large cohorts. The algorithm can find all clusters of ROH regions based on the given parameters:

the minimum number of individuals, the minimum length of the ROH regions, and the objective function. The running time of the algorithm is linear to the size of the genotype panel which enables fast processing of millions of individuals without requiring extravagant resources.

Using ROH-DICE, we conducted a systematic investigation of ROH diplotype clusters in a large population cohort, the UK Biobank. To the best of our knowledge, there has been no such investigation of the genomic distribution of ROH diplotypes conducted previously. We found over 1.8 million ROH diplotype clusters spanning over 100 SNPs and shared by over 100 individuals. While we reported this single data point, the interpretation of the genome-wide ROH diplotype distribution is difficult. First, the expected distribution of ROH diplotype clusters is not known. For populations with an idealized infinite population size, ROH diplotype distribution can be estimated from the allele frequency spectrum. However, for any finite population, when we are looking at haplotypes spanning 100 sites, only a small fraction of possible allele combinations is observed and the distribution will be heavily dependent on the population history. Large ROH clusters can be used to identify signatures of selection in humans or other species. Positive selection reduces haplotype diversity, increasing homozygosity around the target locus, resulting in higher frequencies of ROH in regions containing selection loci (*Pemberton et al., 2012*; *Sabeti et al., 2002*). Therefore, excessive ROH regions can be linked to selective sweeps and have been found to coincide with positive selection in humans (*Pemberton et al., 2012*), and other species (*Hewett et al., 2023*). Three large ROH clusters in chromosomes 2, 6, and 8 of UKBB overlap with known hotspots for selection signals. It should be noted that although the selection of 100 individuals and 100 sites has been used in other studies, it is somewhat arbitrary. While we believe that small variations in the values would not affect the results, using different values such as 200 or 1000 may lead to different ROH clusters. Our preliminary analysis indicates that increasing the length and width of the clusters improves accuracy but reduces power. Future works may investigate the effect of different parameters on the distribution of ROH clusters and downstream analysis.

We found a strong association between non-cancerous diseases and some ROH diplotypes. The majority of ROH regions harboring strong associations with non-cancerous diseases were located in the extended HLA region in chromosome 6. As expected, most of the related diseases were also autoimmune system disorders. While the association signals we found mostly overlap with existing GWAS hits, we are testing different genetic effects. The existing GWAS are mainly testing the additive effects of single SNPs, while we are testing the recessive effects of relatively long haplotypes. In a sense, our analysis is similar to traditional family-based homozygosity mapping (*Lander and Botstein, 1987*), but at a population scale. Future works are warranted to fully develop this potential new gene mapping approach. We want to clarify that we are not claiming ROH-DICE to be superior to regular GWAS in all scenarios. Our simulation only demonstrates that ROH-DICE performs better under certain conditions. Specifically, when the causal variant is located in a long ROH diplotype shared by many individuals (ROH diplotype clusters), ROH-DICE outperforms regular GWAS. It is important to note that ROH-DICE is not meant to replace regular GWAS, but to complement it.

The disease associations presented in this work largely do not represent novel discoveries. The significant associations can be identified in the first place if a recessive mode of inheritance is assumed or a more powerful imputation panel is implemented. However, there is no guarantee that the sites are well imputed if the LD between the genotyped sites is low. We also showed in our simulation that the ROH clusters would outperform GWAS with an additive or even recessive model in terms of power if a phenotype is associated with a set of consecutive homozygous sites.

We used age, gender, and genetic principal components as confounding variables in the association analysis. Genetic principal components can reduce the confounding effect brought on by population structure but it may be insufficient to completely eliminate the effects of recent demographic structure and the local environment (*Zaidi and Mathieson, 2020*). For example, individuals sharing excessive ROH diplotypes may share similar environments since they are closely related and reside close to one another. Since we did not rule out related individuals, some of the reported GWAS signals may not be attributable to ROH.

Our association analysis is a proof of concept and opens up many future opportunities. With our methods, it is possible to extend this analysis to non-disease complex traits. For example, one can investigate whether individuals who share more ROH diplotype clusters have similar phenotypes. Such an analysis may reveal the contribution of dominance variance to the heritability of traits of interest. It

will also be interesting to compare the findings with previous research based on genome-wide aggregate ROH content.

## Methods
### Identification of haplotype clusters in PBWT

The PBWT proposed by *Durbin, 2014* facilitates an efficient approach to search for all pairs of long matches in haplotype or genotype panels. The basic idea behind the PBWT search is to sort the panels at each site by their reversed prefix order. As a result, the matches in the panel will be placed adjacent to each other. However, at the time we started this project, all existing PBWT algorithms (*Durbin, 2014*; *Naseri et al., 2019c*; *Naseri et al., 2019a*; *Sanaullah et al., 2020*) were aimed at identifying pairwise matches. In this work, we propose to employ the PBWT data structures to search for clusters of multi-way matches instead of individual pairs of matches. Independent of our work, a couple of algorithms have been proposed to find haplotype blocks in a PBWT panel (*Cunha et al., 2018*; *Alanko et al., 2020*). The algorithm by *Cunha et al., 2018*; *Alanko et al., 2020* may not be feasible to handle biobank scale data. The recently proposed algorithms by *Cunha et al., 2018*; *Alanko et al., 2020*, however, will scale well for large-scale data, but they aim to enumerate all maximal haplotype blocks. For datasets comprising hundreds of thousands or millions of individuals, the number of reported clusters of any length may be excessive. Moreover, a minimum length threshold in terms of both sites and number of individuals would be more meaningful for downstream analysis especially association analysis, for example where a minimum number of cases are required. Hence, after detecting all possible clusters, filtering has to be applied to remove spurious clusters. Here, we formulate the haplotype blocks problem with two distinct objective functions which will reduce the complexity of filtering the detected clusters afterward.

### Block maximal match problems

Based on the different formulations of the problem, we may have different objective functions: the first problem is to find all clusters with at least $L$ sites that are shared among at least $t$ sequences while

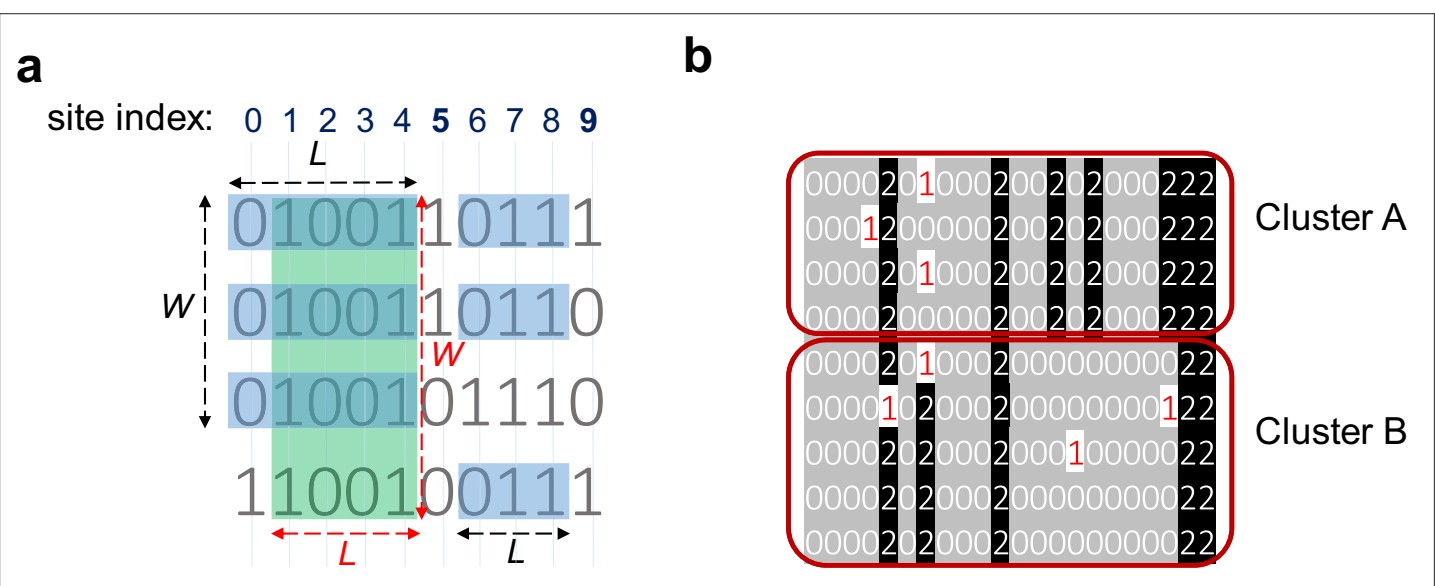

**Figure 6.** Consensuses of haplotype matches with a minimum length ($L$) of 3 and a minimum width ($W$) of 3. (**a**) Clusters of haplotypes with two different objectives: maximizing the number of sites and maximizing the number of individuals. The green rectangle ending at site 4 highlights a cluster that meets the requirement of $W \geq 3$ and $L \geq 3$ while maximizing the number of individuals (width-maximal). The blue rectangles ending at 4 maximize the number of sites (length-maximal). The blue rectangles ending at site 8 show a cluster with $W \geq 3$ and $L \geq 3$ maximizing the number of sites and number of individuals. This cluster is length-maximal because adding either column 5 or 9 will introduce a mismatch; It is also width-maximal because adding the third haplotype will introduce a mismatch. (**b**) Two clusters (clusters A and B) with the same starting and ending positions but different consensuses. Therefore, these two clusters are not merged and considered as separate clusters. Each line represents one individual and 0/0 alleles are highlighted in gray, and 1/1 alleles in black.

maximizing the number of sequences for each cluster. Using proper data structures, we can keep track of the starting position of the matches and report them efficiently. The second problem is to find clusters with at least $L$ sites among at least $t$ sequences while maximizing the number of sites for each cluster. Again, the sequences that share a consensus are put in the same block.

PBWT sorting at the site $k$ places sequences with identical reverse prefixes into clusters of matches that are adjacent to each other. We refer to these clusters as blocks, where the number of sequences $W$ is the width of the block in terms of the number of haplotypes, and the length of matches $L$ is the length of the block in terms of the number of sites. Recall the concept of the set maximal match of *Durbin, 2014* as the pairwise haplotype match that cannot be extended at either end. We extend the concept of set maximal match to *block maximal match*, that is, the haplotype match block that cannot be extended. As the block is a 2D object, the extension can be defined either lengthwise or widthwise. Therefore, we can define the *lengthwise block maximal match* as the matching block that cannot be extended lengthwise. Similarly, the *widthwise block maximal match* is that which cannot be extended widthwise.

For the PBWT block match problem, the goal is to identify all block maximal matches that have a minimal sequence length $L$ and a minimal width $W$. Note that for an identified PBWT block, the boundary of the block may not be exactly defined (see *Figure 6* for an example). We can either report the block boundary that maximizes the length – *length-maximal PBWT block*, or the block boundary that maximizes the width – *width-maximal PBWT block*. We developed exact algorithms for identifying and reporting block maximal matches. This is achieved by using proper data structures tracking the starting position of the matches and the upper and lower boundaries of each matching block. A detailed description of the algorithms is provided in the cPBWT algorithms subsection.

## cPBWT algorithms

### Maximizing the number of haplotypes

Given a haplotype or genotype panel, the objective is to find all matches greater than a given length $L$ that are shared among at least $c$ haplotypes (or individuals). By sorting the panel at each site the matches are placed in the same block. The divergence value for each sequence contains the starting position of the match to its preceding sequence in the reversed prefix order. The matches are separated by a sequence with a divergence value greater than $k − L$. To maximize the number of sequences, the maximum value of the divergence values in each block is considered. The size of the block should also be greater than $c$ to be reported. Algorithm 1 (*Supplementary file 6*) illustrates the procedure for finding long matches while maximizing the number of haplotypes or sequences in detail. Algorithm 2 (*Supplementary file 6*) illustrates the procedure for updating the intermediate variables $V$ and $Q$ to compute $d_{k+1}$ and $a_{k+1}$ based on the $d_k$ and $a_k$. The time complexity of this algorithm is $O(NM)$, where $N$ denotes the number of variant sites and $M$ denotes the number of individuals. Divergence values and prefix arrays are computed in linear time for each variant site and the maximal number of matching blocks at each site is bound by $O(M)$.

### Maximizing the length of the match

The objective is to find the longest matches greater than a given length $L$ shared among at least $c$ sequences. The match will not be reported if the block of matches can be extended while at least $c$ sequences are not terminating. To do this, two conditions should be held: First, at least $c$ sequence for one allele should be present in the block, and second, the $c$th lowest divergence value in the block should be greater or equal to the $c$th lowest divergence of the matches ending with the allele with at least $c$ occurrences. To find the $c$th lowest divergence value, the Quickselect algorithm, a modified one-sided version of Quicksort (*Hoare, 1961*), is used. Quickselect has the average time complexity of $O(N)$, where $N$ denotes the size of the given list. Algorithm 3 (*Supplementary file 6*) illustrates the procedure of finding long matches while maximizing the length of the match in detail.

## ROH-DICE algorithm

Any of the two cPBWT algorithms can be applied to search for ROH–diplotype clusters from genotype data. Maximizing the number of haplotypes would guarantee the inclusion of all samples that may share specific ROH diplotypes. Hence, for association analysis between ROH diplotype and phenotypes, this optimization would be preferred. On the other hand, maximizing the number

of sites would ensure the inclusion of all variant sites between the individuals contributing to the matches which may be more appropriate for other applications such as studying population structures or imputation.

ROH-DICE maps the genotype sequence $x$, defined over the alphabet of {0,1,2}, into a compressed haplotype sequence $y$, defined over the alphabet of {0,1}. For homozygous sites, the mapping is straightforward: for $x_i = 0$, $y_i = 0$; for $x_i = 2$, $y_i = 1$. For heterozygous sites $x_i = 1$, a random value from 0 and 1 was assigned with a probability of ½ for 0 and ½ for 1. The identified maximal matching blocks in the PBWT panel comprising all compressed haplotype sequences $\{y_i\}$, correspond to the approximate ROH clusters in the original genotype sequences $\{x_i\}$. After finding all ROH clusters for a given cut-off, the clusters with the identical start and end positions, and consensus (determined by majority alleles) was merged.

## Identification of ROH hotspots and coldspots

The frequency of ROH calculated over all three size classes at each SNP in the combined data set from the Pemberton study was downloaded (*Pemberton et al., 2012*). The genomic locations were lifted over to hg19 using the liftOver tool (*Hinrichs et al., 2006*). The overlapping ROH cluster from ROH-DICE results with the maximum number of individuals (samples) was assigned for each SNP. ROH hotspots were considered locations where the number of samples in ROH clusters exceeded the 99.5th percentile. ROH coldspots were considered locations where the number of samples in ROH clusters was lower than the 0.5th percentile (equal to 0).

## UK Biobank dataset

The phased haplotype data of the UKBB data (version 2) comprising 658,720 sites were extracted. The Data-Field 20002 contains self-reported non-cancer illnesses comprising 445 categories (diseases). For the association analysis, 430,437 individuals of British ethnicity were selected. The ethnic backgrounds were extracted using the Data-Field 21000.

## Genetic association analysis

We computed the p-values for each disease in all detected ROH clusters that were present in at least 10 individuals. p-values were computed using chi-squared test considering the following numbers: $D1$: Number of individuals sharing a disease within the detected consensus of ROH. $N1$: Number of individuals in the detected ROH not sharing the disease. $D2$: Total number of individuals sharing the disease subtracting $D1$. $N2$: $M − N1 − N2 − D2$, where $M$ denotes the total number of individuals. 100 regions with the lowest p-values (for any disease) were selected and further investigated using PHESANT (downloaded on August 22, 2018).

For PHESANT analysis, age was calculated manually using the date of attending the assessment center (53), year of birth (34), and month of birth (52). Sex (31), genetic principal components (22009), number of self-reported non-cancer illnesses (135), genotype measurement batch (22000), and non-cancer illness (20002) fields were also maintained. PHESANT tests the associations of a trait of interest with a set of other phenotypes, and we considered all diplotypes in the 100 regions as traits of interest. Most of the regions include multiple clusters with the same starting and ending positions but different consensus. We considered all of the clusters in the same region as traits of interest (660 traits of interest in total). Regressions were performed on each diplotype cluster separately, so more than one cluster may have been tested in the same region.

## Retrieval and annotation using the genetic association result from Neale Lab

Each of the associations (computed by PHESANT) was validated against the GWAS results published by Neale's lab [http://www.nealelab.is/blog/2017/9/15/heritability-of-2000-traits-and-disorders-in-the-uk-biobank, accessed July 27, 2018]. For each disease in each cluster (according to PHESANT), all reported SNPs within the genomic region of the cluster that were reported to be associated with the disease (according to Neale's lab results) were searched and the SNP with the lowest p-value was reported.

## Linkage pattern analysis between GWAS and ROH-DICE results

In linkage disequilibrium analysis, $D$ and $D'$ are commonly used measures to quantify the degree of non-random association between alleles at different loci. $D$ measures the difference between the observed frequency of a haplotype and the frequency expected under random mating, while $D'$ is a normalized measure of $D$ that considers the allele frequencies at each locus. In this study, we have adapted these measures between two loci into a location and an ROH cluster.

$D'$ between an ROH cluster and an SNP overlapping the cluster was calculated by normalizing the $D$ between the ROH cluster membership and alternate allele of the SNP similar to linkage analysis between variant sites. Assume $p_R$ is the frequency of samples that belong to the cluster, $p_S$ is the frequency of alternate allele, and $p_{RS}$ is the frequency of samples belonging to the cluster and having the minor allele. We calculate $p_r$ as $1 - p_R$ and $p_s$ as $1 - p_S$. Finally, the $D'$ can be calculated by using the following formula:

$$\text{if } (D < 0) : D' = max\left(-p_R p_S, -p_r p_s\right)$$
$$\text{else: } D' = max\left(p_R p_S, p_r p_s\right)$$
$$\text{where } D = p_{RS} - p_R p_S.$$

## COVID-19 mortality and ROH diplotypes

Two tables 'covid19_result.txt' and 'death.txt' provided by the UK Biobank were downloaded on July 24, 2020. The table 'covid19_result.txt' contains the test results whether the sample was reported as positive or negative for COVID-19. The table 'death.txt' includes the date of death for samples. In the July 24, 2020 release of the table in UK Biobank, 201 British individuals have been reported COVID-19 positive and died in 2020. Those individuals were considered as cases for mortality analysis. A total of 8120 British individuals have been tested for COVID-19. The controls contained the individuals who had been tested but no death information was provided for them. We tested all detected ROH diplotypes for COVID-19 mortality association (with at least 10 cases) using the chi-square test. For the chi-square test, the total number of individuals $M$ corresponds to the number of tested individuals for COVID-19 (8120). GMMAT (*Hoare, 1961*) was used to recalculate the p-value for the diplotype with the lowest p-value from the chi-square test (chr4:106318456–106483898) while adjusting for age, gender, and genomic relationship matrix (GRM). The GRM was computed using the kinship coefficients calculated from KING (*Hoare, 1961*).

## Acknowledgements

AN, SZ, and DZ were supported by the National Institutes of Health grants R01 HG010086 and R56 HG011509. AN and DZ were also supported by the National Institutes of Health grant OT2 OD002751. We thank Dr. Irmgard Willcockson for proofreading.

## Additional information

### Funding

| Funder | Grant reference number | Author |
| --- | --- | --- |
| National Institutes of Health | R01 HG010086 | Ardalan Naseri Degui Zhi Shaojie Zhang |
| National Institutes of Health | R56 HG011509 | Ardalan Naseri Degui Zhi Shaojie Zhang |
| National Institutes of Health | OT2 OD002751 | Ardalan Naseri Degui Zhi |

The funders had no role in study design, data collection, and interpretation, or the decision to submit the work for publication.

## Author contributions

Ardalan Naseri, Conceptualization, Data curation, Software, Validation, Investigation, Visualization, Methodology, Writing – original draft; Degui Zhi, Shaojie Zhang, Conceptualization, Resources, Supervision, Funding acquisition, Methodology, Writing – original draft

## Author ORCIDs

Ardalan Naseri ⓘ https://orcid.org/0000-0002-2747-2193
Degui Zhi ⓘ http://orcid.org/0000-0001-7754-1890
Shaojie Zhang ⓘ http://orcid.org/0000-0002-4051-5549

## Ethics

Our analysis was approved by The University of Texas Health Science Center at Houston committee for the protection of human subjects under No. HSC-SBMI-23-0583. UK Biobank (UKBB) has secured informed consent from the participants in the use of their data for approved research projects. UKBB data were accessed via approved project 24247.

## Decision letter and Author response

Decision letter https://doi.org/10.7554/eLife.81698.sa1
Author response https://doi.org/10.7554/eLife.81698.sa2

# Additional files

## Supplementary files

• Supplementary file 1. Number of individuals in detected runs-of-homozygosity (ROH) clusters in autosomal chromosomes of UKBB.

• Supplementary file 2. The overlapping hotspots between runs-of-homozygosity (ROH)-DICE and *Pemberton et al., 2012*.

• Supplementary file 3. The overlapping coldspots between runs-of-homozygosity (ROH)-DICE and *Pemberton et al., 2012*.

• Supplementary file 4. 100 clusters of the runs-of-homozygosity (ROH) diplotypes with the lowest p-values for self-reported non-cancerous diseases using the British population in UK Biobank.

• Supplementary file 5. Runs-of-homozygosity (ROH) diplotype consensuses of the clusters with the lowest p-values.

• Supplementary file 6. cPBWT algorithms for finding width- and length-maximal matches.

• MDAR checklist

## Data availability

This research has been conducted using the UK Biobank Resource (*Bycroft et al., 2018*) under Application Number 24247. The source code is available at https://github.com/ZhiGroup/ROH-DICE (copy archived at *Naseri, 2024*).

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
