## [Editor Report]

This important study presents a new method for homozygosity mapping in population-scale datasets, based on an innovative computational algorithm that efficiently identifies runs-of-homozygosity (ROH) segments shared by many individuals. Simulation results provided convincing evidence for good accuracy and power of the new algorithm. Application of this new method to the UK Biobank dataset largely recapitulated previously known associations but also revealed a small number of novel discoveries that were missed by existing genome-wide association study methods, highlighting the utility of this new approach. This study will be of substantial interest to readers in human genetics and quantitative genetics.

---

## [Decision Letter]

**Decision letter after peer review:**

Thank you for submitting your article "Discovery of runs-of-homozygosity diplotype clusters and their associations with diseases in UK Biobank" for consideration by *eLife*. Your article has been reviewed by 3 peer reviewers, one of whom is a member of our Board of Reviewing Editors, and the evaluation has been overseen by Molly Przeworski as the Senior Editor. The following individual involved in the review of your submission has agreed to reveal their identity: Shai Carmi (Reviewer #3).

Essential revisions (for the authors):

All three reviewers agree that the approach presented is innovative and potentially useful for future applications in large biobank datasets. However, the performance of the algorithm for identifying ROH diplotype clusters is not quantitatively characterized to demonstrate a low false positive rate. Additionally, the reviewers noted that most of the ROH-phenotype associations identified have already been found by standard GWAS, and more would be found by GWAS with recessive effects considered, which raises the question about the power of the new approach over existing GWAS methods. Lastly, the identified ROH clusters and associations have not been fully reported. Hence, below are the essential revisions that the reviewers suggest, and specific analyses are laid out in detail in their reviews below:

1) The authors need to quantify the error rate of all the identified ROH clusters by going back to the diploid genotype data to confirm that (1) each individual in the cluster is truly (nearly) homozygous across the identified ROH segment; (2) individuals in the same ROH cluster do share the same diplotype. It is also recommended to simulate genotypes with errors (or mutations) to characterize the sensitivity and false positive rate of the ROH detection method. These characterizations will help demonstrate the performance of the new approach and justify (or improve) the choice of parameters for defining ROH clusters (i.e., spanning over 100 SNPs and shared by over 100 individuals).

2) When characterizing the false positive rate of ROH cluster detection, particular attention should be paid to the MHC region, as the reviewers have raised considerable concerns regarding the usually large number of signals in this region. Although the high SNP density and complex linkage-disequilibrium patterns may enable the detection of more short haplotypes shared by identity across individuals, these haplotypes are not expected to be more likely in the homozygous state, so more ROH clusters are not necessarily expected. The authors need to (1) verify （or refute) that most of the identified ROH clusters are real based on the full genotype data; (2) characterize and explain the distribution of ROH segment length in terms of physical or genetic distance, and compare it to that of other chromosomes; (3) consider possibilities of potential technical artifacts, either introduced by the "random allele drawing" of the algorithm or already present in the genotype data (e.g., due to cryptic duplication or structural variation).

3) Although the authors listed a couple of examples where the associations are missed by standard additive effect GWAS, many recent GWAS tools provide alternative models (dominant, recessive, or general) for detecting signals with non-additive effects. It is thus important to quantify the power of the ROH-based method vs. standard (single variant) GWAS with a recessive effect considered.

4) The authors should include more comprehensive reports of the identified ROH clusters and associations in the supplemental materials, which will enable other authors to reproduce the results and carry out follow-up studies.

*Reviewer #1 (Recommendations for the authors):*

Assuming the ROH diplotype cluster identification is accurate, I wonder if the authors could further utilize the identified clusters to explore the genetic architecture of disease risks (or of other complex traits). For example, one can ask if individuals who shared more ROH diplotype clusters tend to be more similar in phenotypes. Such analysis may shed light on the contribution of dominance variance to heritability for traits of interest.

Assuming the ROH association has good power and a low false positive rate, it should be relatively straightforward and of broad interest to extend this analysis to non-disease complex traits. It will also be interesting to compare the results with findings from previous research based on genome-wide aggregate ROH content.

The direction of the ROH association should be reported for each signal in all tables (including supplementary ones) to indicate if homozygosity of certain haplotypes is associated with increased or decreased risk. Similarly, the direction of the effect size of the non-reference allele should be annotated for GWAS results. The linkage pattern between the non-reference allele of GWAS SNP and the ROH segment should be added.

*Reviewer #2 (Recommendations for the authors):*

The algorithm itself is reasonable, however, my biggest concern is that assessment of whether the identified segments are statistically significant is lacking. The authors mentioned "the rate of false positives should be low". However, it is not obvious, and the results should be more specific and quantitative.

First, given the IBD sharing or genetic relatedness in a group of individuals, would the identified ROH be explained by chance alone? This should be evaluated.

Second, what part does linkage disequilibrium play in ROH? As reported, the MHC region of chr6 has a ROH hotspot. The MHC region is known to have an extremely high level of LD. If taken LD level into account, would the ROH clusters be significant?

Third, if SNPs are pre-processed, for example only SNPs with certain MAF and with certain LD distance are kept, how would the results look like? Also, the numbers 100 markers and 100 individuals are quite arbitrary. How would the results depend on the choice of parameters, say with 50 and 200 markers, 50 and 200 individuals?

Fourth, the definition of ROH clusters, blocks, and hotspots should be clearly described.

Fifth, two optimization rules are mentioned, and for the UKB data, only the width-maximal blocks were reported. In practice, what are the criteria to choose one rule over another?

Sixth, the disease associations discussed do not represent new discoveries. The significant associations can be identified in the first place if a recessive mode of inheritance is assumed or a more powerful imputation panel was implemented.

*Reviewer #3 (Recommendations for the authors):*

Please see the annotated PDF file. No need to respond to corrections of typos etc.

[Editors' note: further revisions were suggested prior to acceptance, as described below.]

Thank you for resubmitting your work entitled "Discovery of runs-of-homozygosity diplotype clusters and their associations with diseases in UK Biobank" for further consideration by *eLife*. Your revised article has been evaluated by Detlef Weigel (Senior Editor) and a Reviewing Editor.

The manuscript has been improved but there are some remaining issues that need to be addressed, as outlined below:

1. The newly added evaluation of accuracy and power of ROH diplotype clusters detected by ROH-DICE is appreciated, but this evaluation is based on simulated genotypes of 200 individuals only, and the combination of detection thresholds evaluated do not match those used in the empirical study of UK Biobank data (L=100, W=100). Therefore, it is highly unclear how accurate and powerful ROH-DICE is expected to be in practice. Understandably, simulating a dataset as large as the UK Biobank is infeasible. It will be useful if the authors could provide some back-of-envelope calculation or semi-quantitative estimation of the power in large-scale genomic datasets such as UK Biobank (even an estimate of the order of magnitude would be helpful).

2. The authors argue that the accuracy of ROH diplotype clusters detected in MHC is comparable to other parts of the genome, because there is no excess clusters detected in MHC, when a genetic distance threshold is used instead of the number of consecutive SNPs threshold. However, Figure 3—figure supplement 1 only shows no excess clusters on chromosome 6, without providing specific information regarding the MHC region. Moreover, beyond the total number of clusters, the authors need to show that the size distribution of the ROH diplotype clusters (i.e., number of individuals in each cluster) of MHC is comparable to elsewhere in the genome, as higher SNP density and low recombination rate is not expected to lead to more people sharing the same diplotype. (Related to this point, additional legend/explanation is needed to explain the y-axis, blocks, and colors of Figure 3B, as the current legend is not sufficiently informative.)

3. The evaluation of the power of ROH diplotype association is based on the strong assumption that the causal variant indeed lies in a long, ROH diplotype shared by many individuals (i.e., ROH diplotype clusters). One might argue that the comparison between standard GWAS and ROH-DICE based on this assumption is unfair, because only a small fraction of causal variants reside in ROH diplotype clusters, and standard GWAS may have better power in other scenarios. A more comprehensive and fair comparison is to assign causal variants in the simulated datasets at random, assuming additive, dominant or recessive effect, respectively. The authors can then quantify the frequency of different scenarios (e.g., causal variants in ROH-diplotype clusters of various sizes or not in ROH segments at all) and compare the performance of the two association methods in different scenarios. (Related to this point, more information (such as the simulated sample size, detection thresholds, assignment of causal alleles) is needed in the legend of Figure 1—figure supplement 3.)

---

## [Author Response]

Essential revisions (for the authors):All three reviewers agree that the approach presented is innovative and potentially useful for future applications in large biobank datasets. However, the performance of the algorithm for identifying ROH diplotype clusters is not quantitatively characterized to demonstrate a low false positive rate. Additionally, the reviewers noted that most of the ROH-phenotype associations identified have already been found by standard GWAS, and more would be found by GWAS with recessive effects considered, which raises the question about the power of the new approach over existing GWAS methods. Lastly, the identified ROH clusters and associations have not been fully reported. Hence, below are the essential revisions that the reviewers suggest, and specific analyses are laid out in detail in their reviews below:1) The authors need to quantify the error rate of all the identified ROH clusters by going back to the diploid genotype data to confirm that (1) each individual in the cluster is truly (nearly) homozygous across the identified ROH segment; (2) individuals in the same ROH cluster do share the same diplotype. It is also recommended to simulate genotypes with errors (or mutations) to characterize the sensitivity and false positive rate of the ROH detection method. These characterizations will help demonstrate the performance of the new approach and justify (or improve) the choice of parameters for defining ROH clusters (i.e., spanning over 100 SNPs and shared by over 100 individuals).

1) We filtered out the diplotypes with more than 1% of heterozygous sites. Hence, the number of individuals with nearly homozygous sites should remain the same.

2) Due to the low heterogeneous sites for each individual, most individuals would share the same (similar) consensus. However, as requested, we have conducted additional extensive experiments to evaluate the quality of reported ROH clusters. A new section has been added to the manuscript (Results, Evaluation of ROH clusters in simulated data).

2) When characterizing the false positive rate of ROH cluster detection, particular attention should be paid to the MHC region, as the reviewers have raised considerable concerns regarding the usually large number of signals in this region. Although the high SNP density and complex linkage-disequilibrium patterns may enable the detection of more short haplotypes shared by identity across individuals, these haplotypes are not expected to be more likely in the homozygous state, so more ROH clusters are not necessarily expected. The authors need to (1) verify （or refute) that most of the identified ROH clusters are real based on the full genotype data; (2) characterize and explain the distribution of ROH segment length in terms of physical or genetic distance, and compare it to that of other chromosomes; (3) consider possibilities of potential technical artifacts, either introduced by the "random allele drawing" of the algorithm or already present in the genotype data (e.g., due to cryptic duplication or structural variation).

The MHC region has dense genotype markers and thus the genetic distances between markers for the MHC region grow relatively slower than other regions, which could contribute to the higher number of clusters as the cut-off was in terms of the number of sites. We modified our tool to detect ROH clusters also using the minimum length in genetic distance. We ran ROH-DICE on UK Biobank using L = 0.1 cM and W = 100. The total number of matches is shown in Figure 3—figure supplement 1. As shown in the figure, there is no excessive number of ROH clusters in the MHC region when using genetic distance.

3) Although the authors listed a couple of examples where the associations are missed by standard additive effect GWAS, many recent GWAS tools provide alternative models (dominant, recessive, or general) for detecting signals with non-additive effects. It is thus important to quantify the power of the ROH-based method vs. standard (single variant) GWAS with a recessive effect considered.

We appreciated the suggestion and included a benchmarking for the detection power of ROH-DICE vs. GWAS using dominant, recessive, and additive effects (see the new subsection in Results, Power of ROH-DICE in association studies). Our simulation shows that using ROH clusters outperforms GWAS when a phenotype is associated with a set of consecutive homozygous sites.

4) The authors should include more comprehensive reports of the identified ROH clusters and associations in the supplemental materials, which will enable other authors to reproduce the results and carry out follow-up studies.

More comprehensive information on the reported clusters was added to the supplementary materials. We included the SNP IDs, positions, and even consensus alleles for all reported loci in the main tables. Moreover, additional information including β and D’ values were added. The current information should allow researchers to follow up on the results.

Reviewer #1 (Recommendations for the authors):Assuming the ROH diplotype cluster identification is accurate, I wonder if the authors could further utilize the identified clusters to explore the genetic architecture of disease risks (or of other complex traits). For example, one can ask if individuals who shared more ROH diplotype clusters tend to be more similar in phenotypes. Such analysis may shed light on the contribution of dominance variance to heritability for traits of interest.

This is a great idea. However, our current work aims to achieve three goals: (i) developing an algorithm for identifying ROH clusters, (ii) demonstrating the abundance of ROH clusters in the UK Biobank using the algorithm, and (iii) conducting a preliminary association study of ROH-clusters with diseases. Our work opens up many future opportunities. For instance, one can explore whether individuals who share more ROH diplotype clusters have similar phenotypes. Such an analysis may reveal the contribution of dominance variance to the heritability of traits of interest. We included all three points in the Discussion section and added the new future opportunities at the end of the section.

Assuming the ROH association has good power and a low false positive rate, it should be relatively straightforward and of broad interest to extend this analysis to non-disease complex traits. It will also be interesting to compare the results with findings from previous research based on genome-wide aggregate ROH content.

This is another great point. As we mentioned in the response to the previous question. Our association analysis is only a proof of concept. There should be many opportunities for downstream analysis. We incorporated this idea into our revised Discussion section.

“Our association analysis is a proof of concept and opens up many future opportunities. With our methods, it is possible to extend this analysis to non-disease complex traits. For example, one can investigate whether individuals who share more ROH diplotype clusters have similar phenotypes. Such an analysis may reveal the contribution of dominance variance to the heritability of traits of interest. It will also be interesting to compare the findings with previous research based on genome-wide aggregate ROH content.“

The direction of the ROH association should be reported for each signal in all tables (including supplementary ones) to indicate if homozygosity of certain haplotypes is associated with increased or decreased risk. Similarly, the direction of the effect size of the non-reference allele should be annotated for GWAS results. The linkage pattern between the non-reference allele of GWAS SNP and the ROH segment should be added.

We have added β values for all GWAS and ROH results in Tables 1 and 2 and confirmed that all our ROH-diplotype has a disease-causing effect.

Also, we used D’ as a measure of linkage between the reported GWAS results and ROH clusters. We found that most of the GWAS results and ROH clusters are strongly correlated. However, there are a few cases where D' is small or close to zero. In such cases, the reported p-value from GWAS was also insignificant, while the ROH cluster indicated a significant association (Tables 1 and 2). We also added the details on the calculation of the linkage patterns in the Methods section (see Linkage pattern analysis between GWAS and ROH-DICE results).

Reviewer #2 (Recommendations for the authors):The algorithm itself is reasonable, however, my biggest concern is that assessment of whether the identified segments are statistically significant is lacking. The authors mentioned "the rate of false positives should be low". However, it is not obvious, and the results should be more specific and quantitative.First, given the IBD sharing or genetic relatedness in a group of individuals, would the identified ROH be explained by chance alone? This should be evaluated.

We evaluated the reported ROH clusters and added a new subsection to our Results (see Evaluation of ROH clusters in simulated data subsection). We also did not claim the identified ROH clusters are all statistically significant. The null distribution of the ROH cluster above a certain size LxW is not known and requires statistical modeling of its own. Our contribution is to provide an efficient algorithm for revealing ROH clusters in large cohorts.

Second, what part does linkage disequilibrium play in ROH? As reported, the MHC region of chr6 has a ROH hotspot. The MHC region is known to have an extremely high level of LD. If taken LD level into account, would the ROH clusters be significant?

Our initial method does not consider linkage disequilibrium while calling ROH clusters. The genetic lengths of the reported cluster in the MHC region are smaller due to the high level of LD as mentioned by the reviewer. We now have reported the genetic lengths for the clusters. Setting a cut-off in centiMorgan would decrease the number of ROH clusters significantly. However, most reported ROH clusters contain SNPs known to have a high association with the reported non-cancerous diseases. We also did not use the term hotspot as we are not claiming any statistical significance of the reported ROH clusters and put all mentions of hotspot in quotation marks in the revised manuscript.

Third, if SNPs are pre-processed, for example only SNPs with certain MAF and with certain LD distance are kept, how would the results look like? Also, the numbers 100 markers and 100 individuals are quite arbitrary. How would the results depend on the choice of parameters, say with 50 and 200 markers, 50 and 200 individuals?

We admit the minimal number of markers 100 and the minimal number of individuals 100 are set arbitrarily. The former, the minimal number of markers, is selected to be large enough so that we would not deal with too many clusters. The longer the ROH segment the more likely it is due to some shared genealogy rather than pure statistical noise.

The main rationale is that we want to choose clusters with large enough individuals so that we can correlate them with phenotypes. Of note, the minimum cut-off of 100 individuals was commonly selected in previous studies:

Moreno-Grau, S., Fernández, M.V., de Rojas, I. et al. Long runs of homozygosity are associated with Alzheimer’s disease. Transl Psychiatry 11, 142 (2021). https://doi.org/10.1038/s41398-020-01145-1Lencz, T. et al. Runs of homozygosity reveal highly penetrant recessive loci in schizophrenia. Proc. Natl Acad. Sci. USA 104, 19942–19947 (2007).Christofidou, P. et al. Runs of homozygosity: association with coronary artery disease and gene expression in monocytes and macrophages. Am. J. Hum. Genet. 97, 228–237 (2015).

We included more details in the Results and Discussion sections:

Results:

“We chose a minimal number of markers that is large enough to avoid an extensive number of clusters. Moreover, the longer the ROH segment, the more likely it is due to shared ancestry rather than statistical noise. Our objective is also to select clusters with a sufficiently large number of individuals to correlate them with phenotypes. It is worth noting that in previous studies, a minimum cut-off of 100 individuals was commonly used^6,8,25.^ On average ~18% of sites are heterozygous, and thus for a pair of 100 sites genotype sequences, there is a very small probability that they will be mapped to the same compressed haplotype. Thus, the rate of false positives should be low. To increase statistical power for downstream association tasks, the width-maximal blocks were reported.”

Discussion:

“It should be noted that although the selection of 100 individuals and 100 sites has been used in other studies, it is somewhat arbitrary. While we believe that small variations in the values would not affect the results, using different values such as 200 or 1000 may lead to different ROH clusters. Our preliminary analysis indicates that increasing the length and width of the clusters improves accuracy but reduces power. Future works may investigate the effect of different parameters on the distribution of ROH clusters and downstream analysis.”

Fourth, the definition of ROH clusters, blocks, and hotspots should be clearly described.

We added the definition of hotspots, ROH clusters, and blocks.

(See ROH diplotypes in UK Biobank subsection, Methods overview)

Hotspot: “ROH hotspots in (*27*) refer to locations where the SNP-wise ROH frequency is the 99.5th percentile among all frequencies.“

We also clarified in the Results section that regions with excessive ROH clusters may be hotspots:

“Genomic regions where we found excessive ROH clusters, especially regions with more individuals than in chromosome 15, are potential “ROH hotspots”^33^ in the British population. However, further investigation may be required to confirm “hotspots” as other factors such as marker density may contribute to excessive clusters in certain regions.”

ROH cluster: “We refer to frequent ROH diplotypes above a certain frequency (set of individuals) and a length as ROH clusters.” For our analysis, we used a frequency of 100 and a length of 100.

Haplotype block: “A haplotype matching block is defined as a sequence of variant sites that have a predefined minimum frequency”

We also clarified in the manuscript (Results section, ROH diplotypes in UK Biobank subsection): “Please note that both definitions of “ROH hotspots” and ROH clusters here are operational and may not indicate any statistical significance.”

Fifth, two optimization rules are mentioned, and for the UKB data, only the width-maximal blocks were reported. In practice, what are the criteria to choose one rule over another?

Maximizing the number of haplotypes would guarantee the inclusion of all samples that may share specific ROH diplotypes. Hence, for association analysis between ROH diplotype and phenotypes, this optimization would be preferred. On the other hand, maximizing the number of sites would ensure the inclusion of all variant sites between the individuals contributing to the matches which may be more appropriate for other applications such as studying population structures or imputation. We included it in the Methods section, ROH-DICE algorithm subsection.

Sixth, the disease associations discussed do not represent new discoveries. The significant associations can be identified in the first place if a recessive mode of inheritance is assumed or a more powerful imputation panel was implemented.

We agree with your comments and discussed these issues in the Discussion section:

“The disease associations presented in this work largely do not represent novel discoveries. The significant associations can be identified in the first place if a recessive mode of inheritance is assumed or a more powerful imputation panel is implemented. However, there is no guarantee that the sites are well-imputed if the LD between the genotyped sites is low. We also showed in our simulation that the ROH clusters would outperform GWAS with an additive or even recessive model in terms of power if a phenotype is associated with a set of consecutive homozygous sites.”

Reviewer #3 (Recommendations for the authors):Please see the annotated PDF file. No need to respond to corrections of typos etc.

We responded to all comments and suggestions in the annotated PDF file.

[Editors’ note: what follows is the authors’ response to the second round of review.]

The manuscript has been improved but there are some remaining issues that need to be addressed, as outlined below:1. The newly added evaluation of accuracy and power of ROH diplotype clusters detected by ROH-DICE is appreciated, but this evaluation is based on simulated genotypes of 200 individuals only, and the combination of detection thresholds evaluated do not match those used in the empirical study of UK Biobank data (L=100, W=100). Therefore, it is highly unclear how accurate and powerful ROH-DICE is expected to be in practice. Understandably, simulating a dataset as large as the UK Biobank is infeasible. It will be useful if the authors could provide some back-of-envelope calculation or semi-quantitative estimation of the power in large-scale genomic datasets such as UK Biobank (even an estimate of the order of magnitude would be helpful).

We simulated a new dataset containing 1000 individuals and added a genotyping error of 0.1%. The ground truth clusters were extracted similarly to the smaller panel. The power for *L* = 100 and *W* = 100 was 55.96% and accuracy 52.84%, while 58.97% of the reported clusters overlap 50% or more with a ground truth cluster. We included the new numbers in the Results section, Evaluation of ROH clusters in simulated data subsection.

2. The authors argue that the accuracy of ROH diplotype clusters detected in MHC is comparable to other parts of the genome, because there is no excess clusters detected in MHC, when a genetic distance threshold is used instead of the number of consecutive SNPs threshold. However, Figure 3—figure supplement 1 only shows no excess clusters on chromosome 6, without providing specific information regarding the MHC region. Moreover, beyond the total number of clusters, the authors need to show that the size distribution of the ROH diplotype clusters (i.e., number of individuals in each cluster) of MHC is comparable to elsewhere in the genome, as higher SNP density and low recombination rate is not expected to lead to more people sharing the same diplotype. (Related to this point, additional legend/explanation is needed to explain the y-axis, blocks, and colors of Figure 3B, as the current legend is not sufficiently informative.)

We have updated Figure 3—figure supplement 1 by replacing it with a new figure that shows the exact number of clusters in the MHC region. The latest figure displays the cluster counts for British people, similar to Figure 3, but after filtering out all clusters smaller than 0.1 cM. The number of clusters in the MHC regions reduced from 65458 to 6813 after the genetic length filtering. Figure 4 shows the size distribution of the clusters, and we have now added a new figure supplement (Figure 4—figure supplement 1) that demonstrates the number of samples in ROH clusters across the chromosomes after filtering. The number of samples in ROH clusters within the MHC regions reduces significantly (see Figure 4—figure supplement 1). Although there is still a peak, it is comparable to other chromosomes such as chromosome 10 or 12.

3. The evaluation of the power of ROH diplotype association is based on the strong assumption that the causal variant indeed lies in a long, ROH diplotype shared by many individuals (i.e., ROH diplotype clusters). One might argue that the comparison between standard GWAS and ROH-DICE based on this assumption is unfair, because only a small fraction of causal variants reside in ROH diplotype clusters, and standard GWAS may have better power in other scenarios. A more comprehensive and fair comparison is to assign causal variants in the simulated datasets at random, assuming additive, dominant or recessive effect, respectively. The authors can then quantify the frequency of different scenarios (e.g., causal variants in ROH-diplotype clusters of various sizes or not in ROH segments at all) and compare the performance of the two association methods in different scenarios. (Related to this point, more information (such as the simulated sample size, detection thresholds, assignment of causal alleles) is needed in the legend of Figure 1—figure supplement 3.)

**“**We want to clarify that we are not claiming ROH-DICE to be superior to regular GWAS in all scenarios. Our simulation only demonstrates that ROH-DICE performs better under certain conditions. Specifically, when the causal variant is located in a long ROH diplotype shared by many individuals (ROH diplotype clusters), ROH-DICE outperforms regular GWAS. It is important to note that ROH-DICE is not meant to replace regular GWAS, but to complement it.”

We included the above paragraph in the Discussion section. We also added the number of samples, chromosome length, and the number of casual variants (100) to the caption of the figure.